# Bridging the gap: Using reservoir ecology and human serosurveys to estimate Lassa virus spillover in West Africa

Andrew J. Basinski[1]*, Elisabeth Fichet-Calvet[2], Anna R. Sjodin[3], Tanner J. Varrelman[4], Christopher H. Remien[1], Nathan C. Layman[3], Brian H. Bird[5], David J. Wolking[5], Corina Monagin[5], Bruno M. Ghersi[5], Peter A. Barry[6], Michael A. Jarvis[7], Paul E. Gessler[8], Scott L. Nuismer[3]

**1** Department of Mathematics, University of Idaho, Moscow, Idaho, United States of America, **2** Department of Virology, Bernhard-Nocht Institute of Tropical Medicine, Hamburg, Germany, **3** Department of Biological Sciences, University of Idaho, Moscow, Idaho, United States of America, **4** Bioinformatics and Computational Biology, University of Idaho, Moscow, Idaho, United States of America, **5** One Health Institute, School of Veterinary Medicine, University of California, Davis, California, United States of America, **6** Center for Comparative Medicine, California National Primate Research Center, Department of Pathology and Laboratory Medicine, University of California, Davis, California, United States of America, **7** School of Biomedical and Healthcare Sciences, University of Plymouth, Plymouth, United Kingdom, **8** College of Natural Resources, University of Idaho, Moscow, Idaho, United States of America

* abasinski@uidaho.edu

**Data Availability Statement:** Our full data-set and the script files used to fit the models are available

## Abstract

Forecasting the risk of pathogen spillover from reservoir populations of wild or domestic animals is essential for the effective deployment of interventions such as wildlife vaccination or culling. Due to the sporadic nature of spillover events and limited availability of data, developing and validating robust, spatially explicit, predictions is challenging. Recent efforts have begun to make progress in this direction by capitalizing on machine learning methodologies. An important weakness of existing approaches, however, is that they generally rely on combining human and reservoir infection data during the training process and thus conflate risk attributable to the prevalence of the pathogen in the reservoir population with the risk attributed to the realized rate of spillover into the human population. Because effective planning of interventions requires that these components of risk be disentangled, we developed a multi-layer machine learning framework that separates these processes. Our approach begins by training models to predict the geographic range of the primary reservoir and the subset of this range in which the pathogen occurs. The spillover risk predicted by the product of these reservoir specific models is then fit to data on realized patterns of historical spillover into the human population. The result is a geographically specific spillover risk forecast that can be easily decomposed and used to guide effective intervention. Applying our method to Lassa virus, a zoonotic pathogen that regularly spills over into the human population across West Africa, results in a model that explains a modest but statistically significant portion of geographic variation in historical patterns of spillover. When combined with a mechanistic mathematical model of infection dynamics, our spillover risk model predicts that 897,700 humans are infected by Lassa

in the github repository: https://github.com/54481andrew/pathogen-spillover-forecast.

**Funding:** Funding was provided by DARPA grant no. D18AC00028 (to BHB, PAB, MAJ, SLN) and NIH grant no. R01GM122079 (to SLN). The funders had no role in study design, data collection and analysis, decision to publish, or preparation of the manuscript.

**Competing interests:** The authors have declared that no competing interests exist.

virus each year across West Africa, with Nigeria accounting for more than half of these human infections.

## Author summary

The 2019 emergence of SARS-CoV-2 is a grim reminder of the threat animal-borne pathogens pose to human health. Even prior to SARS-CoV-2, the spillover of pathogens from animal reservoirs was a persistent problem, with pathogens such as Ebola, Nipah, and Lassa regularly but unpredictably causing outbreaks. Machine-learning models that anticipate when and where pathogen transmission from animals to humans is likely to occur would help guide surveillance efforts and preemptive countermeasures like information campaigns or vaccination programs. We develop a novel machine learning framework that uses datasets describing the distribution of a virus within its host and the range of its animal host, along with data on spatial patterns of human immunity, to infer rates of animal-to-human transmission across a region. By training the model on data from the animal host alone, our framework allows rigorous validation of spillover predictions using human data. We apply our framework to Lassa fever, a viral disease of West Africa that is spread to humans by rodents, and use the predictions to update estimates of Lassa virus infections in humans. Our results suggest that Nigeria is most at risk for the emergence of Lassa virus, and should be prioritized for outbreak-surveillance.

## Introduction

Emerging infectious diseases (EIDs) pose a persistent threat to public health. Approximately 60% of EIDs are caused by pathogens that normally circulate in wild or domestic animal reservoirs (i.e., zoonotic pathogens) [1]. Prior to full scale emergence, interaction between humans and wildlife creates opportunities for the occasional transfer, or spillover, of the zoonotic pathogen into human populations [2]. These initial spillover infections, in turn, represent newly established pathogen populations in human hosts that are subject to evolutionary pressures and may potentially lead to increased transmission among humans [2, 3]. Consequently, a key step in preempting the threat of EIDs is careful monitoring of when and where spillover into the human population occurs. However, because the majority of EIDs from wildlife originate in low and middle income regions with limited disease surveillance, accurately estimating the rate and geographical range of pathogen spillover, and therefore the risk of new EIDs, is a major challenge [1].

Machine learning techniques have shown promise at predicting the geographical range of spillover risk for several zoonotic diseases including Lassa fever [4–6], Ebola [7, 8], and Leishmaniases [9]. Generally, these models are trained to associate environmental features with the presence or absence of case reports in humans or the associated reservoir. Once inferred from the training process, the learned relationships between disease presence and the environment can be extended across a region of interest. Using these techniques, previous studies of Lassa fever (LF) have derived risk maps that assess the likelihood of human LF cases being present in different regions of West Africa [4, 5]. However, because these forecasts combine case-reports from both rodents and humans in the training process, they conflate attributes of the human and reservoir populations that increase spillover risk. Consequently, these approaches shed

little light on aspects of reservoir or human populations that determine the magnitude of spillover at a location and thus miss opportunities to identify effective interventions.

We develop a multi-layer machine learning framework that accounts for the differences between how data involving a wildlife reservoir, and data from human serosurveys, can simultaneously inform spillover risk in people and rigorously assess whether predicted risk quantifies the rate of new infections in humans. Our approach uses machine learning algorithms that, when trained on data from the wildlife reservoir alone, estimate the likelihood that the reservoir and the zoonotic pathogen are present in an area. These predictions are then combined into a composite estimate of spillover risk to humans. Next, our framework uses estimates of human pathogen seroprevalence, as well as estimates of human population density, to translate the composite risk estimate into a prediction of the realized rate of zoonotic spillover into humans. Omitting human seroprevalence data from the training process of the risk-layer has several advantages. First, in the case of LF, due to modern transportation and the longevity of Lassa virus antibodies in humans, a general concern is that the reported location of individual cases of human disease or Lassa virus antibody detection is not the site at which the infection occurred [10–12]. If the dispersal ability of the reservoir is small, training the risk layer on reservoir infections alone helps the model avoid these biases when learning the spatial variation of spillover risk. Secondly, in our framework, human seroprevalence estimates provide an ultimate test of the risk layer's ability to correlate with spatial variation in the cumulative human exposure to the pathogen. The seroprevalence data, in turn, stem from population-based surveys at a site and are therefore much less likely to be influenced by the movement of individuals.

We apply our framework to Lassa virus (formally *Lassa mammarenavirus* [LASV]), a bi-segmented, single-stranded ambisense RNA virus in the *Arenaviridae* family and the causative agent of LF in West Africa [11, 13]. Though LASV can transmit directly between humans and often does so in hospital settings [14], rodent-to-human transmission accounts for the majority of new LASV infections [11, 15]. Specifically, the multimammate rat *Mastomys natalensis* is believed to be responsible for most of the transmission into the human population, either through consumption of food contaminated by rodent feces and urine or through hunting and consumption of the rodent reservoir itself [16]. What remains largely unknown, however, is the extent to which spatial patterns of spillover are driven by spatial variation in the abundance of *M. natalensis* and viral prevalence within *M. natalensis* relative to spatial variation in other contributing factors such as human behavior, housing materials, or other rodent reservoirs. An additional unknown is the true magnitude of spillover into the human population outside of the few areas in Sierra Leone and Nigeria where hospitals with Lassa diagnostic capacity exist. As a consequence, most estimates for the magnitude of Lassa virus spillover rely on longitudinal serosurveys conducted in the 1980s in Sierra Leone [17], yielding estimates of between 100,000 and 300,000 LASV infections each year across West Africa. Here, we use our framework to fill these important gaps in our current understanding of Lassa virus spillover within West Africa.

## Data and study region

We used online data repositories and literature sources to collect three types of data in West Africa spanning the time-range 1970—2017: 1) capture-locations of *M. natalensis*, as well as occurrence locations of non-Mastomys murids; 2) locations and outcomes of LASV surveys conducted in *M. natalensis*; and 3) locations and measured seroprevalence of human LASV serosurveys. The focal region from which our data originate, shown in Fig 1, was chosen as the intersection of West Africa and the International Union for Conservation of Nature (IUCN)

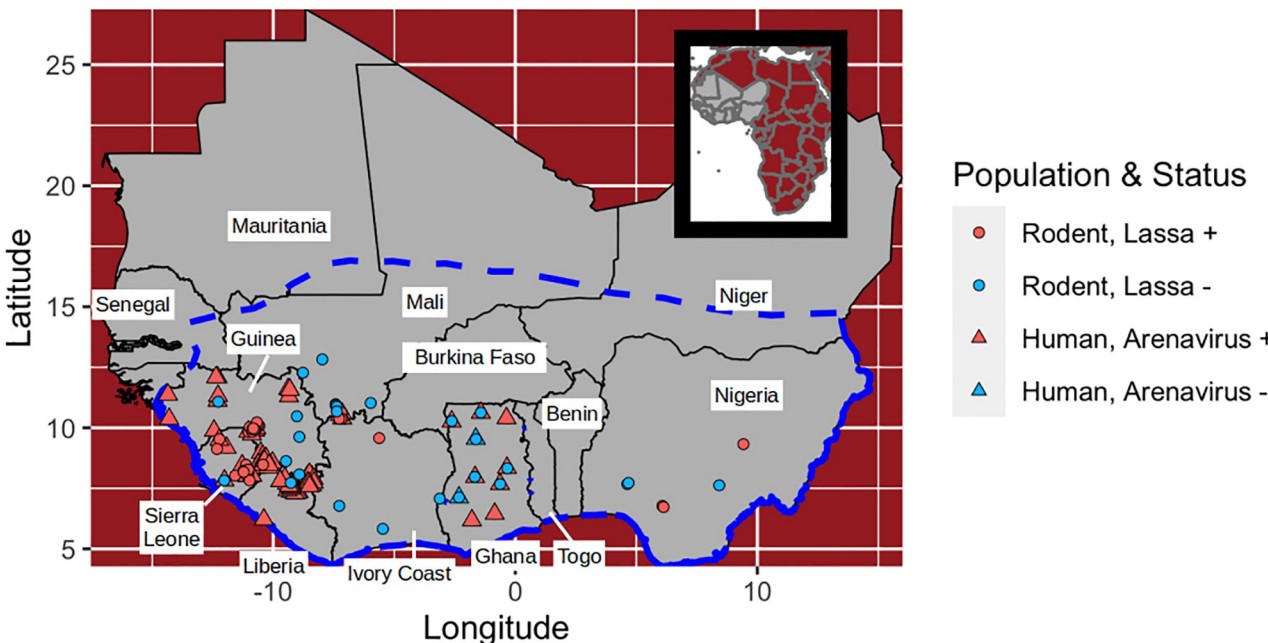

**Fig 1. Map of the study region.** The dashed blue line indicates the study region from which rodent and human data originate. Dots indicate locations at which Lassa virus or arenavirus antibodies have been sampled in rodents or humans. Each rodent point shows the outcome of a serological or PCR test. Each human population point shows the location of a serosurvey.

range map for *Mastomys natalensis* [18, 19]. Though *M. natalensis* is widely distributed across all of Africa, the species consists of multiple clades that likely differ in their ability to serve as hosts to LASV [20, 21]. By limiting the spatial extent of the study region to West Africa we focus on the region occupied by the A-I clade of *M. natalensis* that is believed to transmit LASV [22]. Our *M. natalensis* capture data, as well as all of the LASV survey data, originate from within this region, thus providing a discrete bound on the area of Africa in which the learned relationships of the model apply. For these analyses, this study region was divided into 0.05˚x0.05˚ pixels (approximately 5 km by 5 km at the equator).

The first two datasets generate response variables for the model layers that predict LASV risk. The human seroprevalence data are used to evaluate the combined LASV risk layer for its ability to predict LASV spillover in humans and are also used to calibrate the stage of the model that predicts human LASV spillover. Our full dataset and the script files used to fit the models are available in a github repository [23].

### *Mastomys natalensis* presence data and background

We collected data on historical captures of *M. natalensis* from various sources. First, several sources were used to identify all countries of West Africa that contain *M. natalensis* [24–26]. Next, rodent and mammal databases, as well as literature sources, were cross-referenced to fill in details regarding the year of capture, latitude/longitude coordinates, and the method of identification for each location at which *M. natalensis* was documented [17, 20, 27–42]. Because *M. natalensis* is morphologically similar to other rodents in the study region (e.g., *Mastomys erythroleucus*), we only include those presences that have been confirmed with genetic methods or skull morphology. We found 167 locations with confirmed *M. natalensis* captures. All *M. natalensis* captures occurred in the time-range 1977—2017.

**Table 1. Summary of rodent captures used in the reservoir layer.**

| Country | Year | # Pseudoabsences | # Presences |
|---|---|---|---|
| Benin | 2001-2017 | 12 | 7 |
| Burkina Faso | 1977-2008 | 3 | 15 |
| Ghana | 1999-2011 | 13 | 9 |
| Guinea | 1996-2012 | 71 | 12 |
| Guinea-Bissau | 2013 | 1 | 0 |
| Ivory Coast | 1978-2010 | 21 | 8 |
| Liberia | 1980-2013 | 18 | 0 |
| Mali | 1979-2012 | 58 | 47 |
| Niger | 1977-2007 | 16 | 14 |
| Nigeria | 1977-2015 | 7 | 13 |
| Senegal | 1990-2005 | 0 | 13 |
| Sierra Leone | 1977-2014 | 31 | 17 |
| Togo | 1982 | 1 | 0 |
| **Aggregate** | 1977-2017 | 252 | 155 |

# Pseudoabsences shows the number of unique $0.05 \times 0.05°$ pixels in the GBIF dataset for which only non-Mastomys rodents were captured. # Presences indicates the number of pixels in which one or more *M. natalensis* was captured.

Fitting the model requires supplementing the presence-only data with background points, also called pseudo-absences [43, 44]. Background points serve as an estimate of the distribution of sampling effort for the organism being modeled [45]. We used background points chosen from locations where rodents in the family Muridae had been captured in West Africa from the Global Biodiversity Information Facility (GBIF) website [46]. We filtered the original dataset to reduce the likelihood of including *M. natalensis* rodents that were misidentified as *M. erythroleucus* and vice versa. Namely, we omit from the collection all Murid occurrences that are within the genus *Mastomys*. In addition, to ensure that the GBIF captures are concurrent with captures of *M. natalensis*, we only retained captures that occurred in the time-frame of the *M. natalensis* captures. Finally, we only included records that are within the study region depicted in Fig 1 and that fall outside of any pixel that contains a documented *M. natalensis*. The resulting GBIF dataset spans the years 1977—2015.

These data were used to categorize the subset of the pixels that contained one or more captures into two exclusive categories: those in which at least one *M. natalensis* had been captured (termed presences), and those with only non-Mastomys rodent occurrences (termed pseudo-absences). In total, our dataset classified 155 unique pixels as capture-positive for *M. natalensis*, and 252 pixels as background (Table 1).

## Surveys of *Mastomys natalensis* for Lassa virus

We compiled a dataset that contains occurrences of LASV in rodents or humans. The dataset was established by an extensive review of LASV literature. Primary sources were found by PubMed and GenBank searches of the terms "Lassa", "Lassa fever", "Lassa virus", "Lassa arenavirus", and "Lassa mammarenavirus" [47]. Data from these primary sources was organized into an Excel workbook.

From the full LASV dataset, we collected published studies that sampled *M. natalensis* rodents for indicators of LASV. For each study, we found the sampling location for each tested rodent (either latitude/longitude or a locality name for which coordinates could be obtained).

In total, we compiled thirteen rodent studies [17, 30, 34, 36, 39, 41, 42, 48–53] that tested *M. natalensis* for LASV and contained latitude/longitude coordinates. The resulting test locations originate from six countries and span the years 1972–2014.

Because the prevalence of LASV in rodents varies seasonally [54], and because of the sparsity of time-series data that might otherwise allow the average LASV prevalence in rodents to be estimated, we used the collected data to broadly classify pixels into the categories "Lassa positive" or "Lassa negative". Specifically, a pixel was defined as Lassa positive if, at some point, a *M. natalensis* rodent was captured within the pixel, and the rodent tested positive for LASV using a RT-PCR assay or viral isolation. Because arenavirus antibodies cross-react, a positive LASV antibody test in an individual rodent only indicates past infection with an arenavirus, and not necessarily LASV. In an effort to reduce the frequency of false positives in the training data, pixels that only contain LASV seropositive tests of rodents, and no positive LASV viral detection, were not used as training data. These criteria led to the omission of eight pixels from the training data. Fitting the model with these eight pixels included as presences is an option in the code on the github repository, but does not substantially affect the overall fit of the model [23].

Although serosurveys of rodents cannot specifically show that LASV is present, they can indicate the absence of LASV (along with all other arenaviruses). Pixels were classified as Lassa negative if five or more *M. natalenis* rodents in total were tested for infection with LASV by RT-PCR, or tested for any previous arenavirus exposure using a serological assay, and all rodents tested were negative. We chose a threshold of five to help reduce the chance of including false negatives (i.e. sites that have LASV but in which only non-exposed rodents were captured). This procedure allowed us to classify 62 unique pixels in total: 27 were classified as Lassa negative, and 35 were classified as Lassa positive (Table 2 and Fig 1).

## Human seroprevalence data

From our full LASV dataset described in the previous section, we collected literature sources that describe the prevalence of arenavirus antibodies in human populations of West Africa. As with the rodent LASV infection data, arenavirus antibodies are not specific to LASV. However, because human serosurveys were often conducted in LASV endemic areas or near documented locations of LASV-infected rodents, these serosurveys likely measured the fraction of humans with previous LASV infection, rather than exposure to another arenavirus. We required that each literature source include information on the diagnostic method that was used to test individuals (e.g., ELISA, IFA) and broad details of the survey design. We only

**Table 2. Summary of LASV positive and LASV negative pixels used in the pathogen layer.**

| Country | Year | # Pixels | # Neg. Pixels | # Pos. Pixels |
|---------|------|----------|---------------|---------------|
| Ghana | 2010-2011 | 7 | 7 | 0 |
| Guinea | 2003-2014 | 19 | 6 | 13 |
| Ivory Coast | 2003-2013 | 4 | 3 | 1 |
| Mali | 2004-2012 | 11 | 7 | 4 |
| Nigeria | 1972-2012 | 6 | 3 | 3 |
| Sierra Leone | 1972-2009 | 15 | 1 | 14 |
| **Aggregate** | 1972-2014 | 62 | 27 | 35 |

Each row aggregates literature and GenBank data sources over a country. # Pos. Pixels indicates the number of unique pixels that had one or more LASV-infected rodents. # Neg. Pixels is the number of pixels in which five or more rodents were tested and found negative for LASV infection or antibody.

**Table 3. Summary of human arenavirus serosurveys used in the model.**

| Country | Year | # Sites | Method | # Tested | % Seropositive | Reference |
|---------|------|---------|--------|----------|----------------|-----------|
| Ghana | 2010-2011 | 10 | ELISA | 657 | 5 | [57] |
| Guinea | 2000 | 30 | IFA | 977 | 11 | [55] |
| Guinea | 1990-1993 | 28 | ELISA | 3276 | 23 | [56] |
| Liberia | 1980-1982 | 7 | IFA | 1848 | 5 | [59] |
| Mali | 2015 | 3 | ELISA | 600 | 33 | [58] |
| Sierra Leone | 1977-1983 | 14 | IFA | 5098 | 23 | [17] |
| Sierra Leone | 1970-1972 | 2 | CF | 255 | 6 | [60] |
| **Aggregate** | 1970-2015 | 94 | | 12,711 | 19 | |

Each row is an individual literature source. For each study, # Sites shows the number of locations at which arenavirus surveys were performed, # Tested indicates the total number of individuals tested across sites, and % Seropositive shows the percentage of individuals that tested positive across all sites.

included survey studies that were designed to estimate the seroprevalence in the local community population. This criterion excluded surveys of hospitals, for example, as well as surveys of missionaries.

Each datum contains latitude and longitude of the serosurvey, the number of individuals tested, and the number of individuals determined to have arenavirus antibodies. In total, we collected 94 serosurveys from seven studies (Fig 1) [17, 55–60]. These serosurveys were conducted between 1970 and 2015 and are located in five countries in West Africa (Table 3 and Fig 1).

## Predictors

We include predictors that are broadly hypothesized to influence the distributions of *M. natalensis* and LASV. *M. natalensis* is widely distributed across sub-Saharan Africa in savanna and shrubland environments. Within such environments, *M. natalensis* is commonly associated with small rural communities and is considered a serious agricultural pest [19, 54]. To allow the model the possibility to learn these relationships, we include predictors that describe MODIS land cover features as predictors, and also include human population density within each pixel. We also include elevation in meters. Because climate seasonality and crop maturation affect the breeding season of *M. natalensis*, we include various measures of the seasonality of the vegetative index (NDVI), precipitation, and temperature [61]. See S1 Appendix for a complete list of environmental variables. LASV is often associated with *M. natalensis*, so we use the same set of predictors for the pathogen layer.

## Methods

We developed a model that predicts the rate of LASV infection in humans within individual 0.05˚x0.05˚ pixels across West Africa. An overview of the model framework is depicted in Fig 2. Outputs from the model are generated in two stages. The first stage uses environmental features to estimate different layers of LASV spillover risk. The layers of risk, in turn, are described by: 1) $D_M$, a classification score indicating the likelihood that a pixel contains the primary rodent reservoir, *M. natalensis*, and 2) $D_L$, a score indicating the likelihood that LASV circulates within the *M. natalensis* population, conditioned on the rodent being present. Depending on the layer, the response variable for this stage is generated from documented occurrences of *M. natalensis* ($D_M$ layer), or evidence of past LASV infection in *M. natalensis* ($D_L$ layer). These layers are used to define a composite layer of spillover risk $D_X$, the product of $D_M$ and $D_L$, that describes the likelihood that a pixel simultaneously contains *M. natalensis*

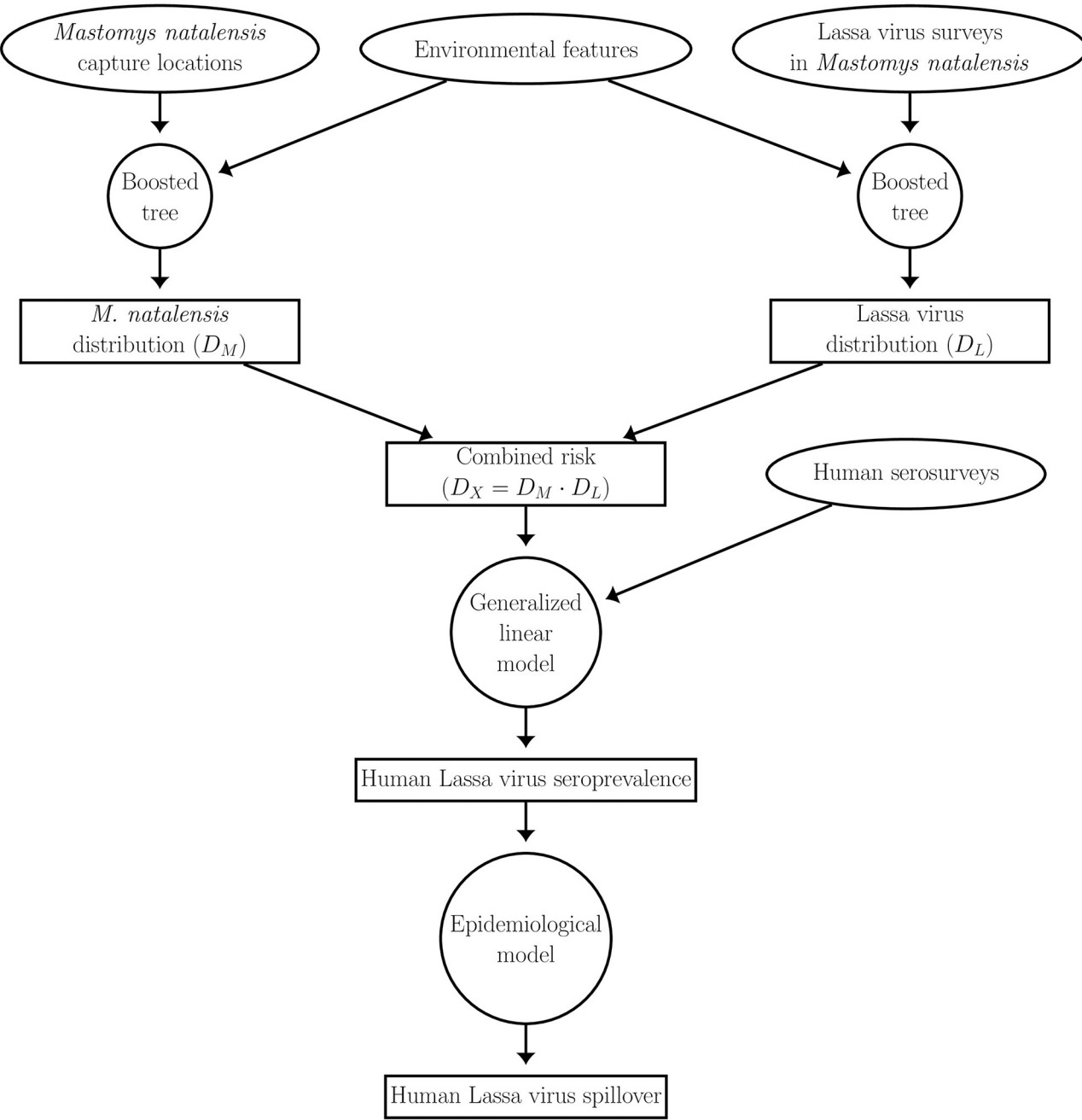

**Fig 2. Overview of the model.** Ellipses represent datasets, circles represent models, and rectangles represent model predictions.

and LASV. The second stage of our framework uses a generalized linear model to regress the estimates of human arenavirus seroprevalence onto the $D_X$ layer. Lastly, we used an epidemiological model to estimate human incidence from the predictions of seroprevalence.

## LASV risk layers

Each risk layer of the first stage is generated by a separate boosted classification tree (BCT). The BCT, in turn, uses environmental features within a pixel to infer a classification score,

between zero and one, that indicates how likely it is that the pixel is positive for *M. natalensis* ($D_M$ layer) or LASV in *M. natalensis* ($D_L$ layer). BCTs use a stage-wise learning algorithm that, at each stage, trains a new tree model to the residuals of the current model iteration. Each newly fitted tree is added to the ensemble model, thereby reducing the residual deviance between the model predictions and a training set [62]. Boosted trees are commonly used in species and disease distribution models because they are simultaneously resistant to over-fitting in scenarios where many feature variables are implemented and are also able to model complex interactions among features [63].

Prior to inclusion in the model-fitting procedure, each feature variable was vetted for its ability to distinguish between presences and absences in each of the layers. Specifically, for each risk layer's binary response variable, we performed a Mann-Whitney U-test on each candidate feature. In doing so, we test the null hypothesis that the distribution of a feature is the same between pixels that are classified as a presence or (pseudo) absence. We only include predictors for which the null hypothesis is rejected at the $\alpha = 0.05$ level.

For a given training set, we fit the BCT model using the gbm.step function of the "dismo" package in the statistical language R [64]. This specific function uses 10-fold cross-validation to determine the number of successive trees that best model the relationship between response and features without over-fitting the data [64]. The learning rate parameter, which determines the weight given to each successive tree, was set to small values ($D_M$: $10^{-2}$, $D_L$: $10^{-3}$) that encourage a final model that is composed of many small incremental improvements. A smaller learning rate was used in the $D_L$ layer because the corresponding dataset was smaller. The parameter that describes the maximum number of allowable trees was set to a large value ($10^7$) to ensure that the cross-validation fitting process was able to add trees until no further improvement occurred [62].

For the $D_M$ layer, we trained 25 boosted classification trees to learn how environmental predictors influence the suitability of a habitat for *M. natalensis*. Each model was fit by selecting 155 presence pixels and pairing these with 155 background pixels in which only non-Mastomys murids were found. Both presences and background pixels were chosen with replacement. By choosing equal numbers of presences and background pixels for each training set, we encourage each model to learn patterns in features that allow presences to be discriminated from background pixels, rather than having the model learn the (likely biased) distribution of presences and background pixels that are available in the overall dataset [44].

For each model fit for the $D_M$ layer, presence and pseudo-absence pixels that were not used to train the model (i.e., out-of-bag data) were used to test the model using the area-under-the-receiver-curve (AUC). The AUC measures a classifier's ability to assign a high classification score to presences, and a low score to background pixels. A score of one indicates a perfect classifier, and a score of 0.5 indicates a classifier that is no better than chance. A pairwise-distance sampling scheme was used to pair an equal number of test-background pixels to the out-of-bag presences that together comprise the test set. Specifically, for each test presence point, the pairwise distance sampling method chooses a test background point so that the minimum spatial distance between the training presences and test presence is similar to the minimum distance between the test background point and training presences [65]. Compared to random selection of test background points, pairwise distance sampling oftentimes results in a lower AUC score that more accurately measures the model's ability to generalize to new regions [65].

The $D_L$ layer is generated by the averaged predictions of 25 boosted classification tree models, each of which is trained to discriminate between pixels that are Lassa positive or Lassa negative. We trained each model on a dataset comprised of 27 absence locations and 27 presence locations, sampled from the full dataset with replacement. The estimation of error in the $D_L$

layer is similar to that described in the $D_M$ layer. Specifically, we calculate the AUC of the fitted model on an equal number of out-of-bag presences and absences.

Next, we combined the $D_M$ and $D_L$ layers into a composite feature, denoted by $D_X$, that is indicative of whether a pixel simultaneously has environmental features that are suitable for *M. natalensis*, as well as LASV in *M. natalensis*. The combined feature is defined as $D_X = D_M \times D_L$ and summarizes the realized risk of LASV spillover to humans within the local environment.

## Connection to human seroprevalence and incidence

To connect the new risk parameter $D_X$ to human arenavirus seroprevalence, and to evaluate the ability of the $D_X$ layer to explain historical LASV spillover in humans, we regressed seroprevalence from human arenavirus serosurveys on the $D_X$ layer and an intercept. In doing so, we test whether human seroprevalence is significantly associated with the probability $D_X$ that a pixel contains LASV-infected *M. natalensis*. We used quasi-binomial regression to account for overdispersion in seroprevalence measurements that could otherwise contaminate hypothesis tests on model coefficients [66]. More details on the motivation behind the quasi-binomial regression can be found in the S1 Appendix. In the regression, each seroprevalence estimate is weighted by the number of individuals tested in the serosurvey.

Next, we used an epidemiological model, based on the classic susceptible-infected-recovered framework, to derive an equation that relates a given LASV spillover rate into humans and the resulting seroprevalence in a human population. Throughout, we assume that the seroprevalence measures that were obtained from historical serosurveys describe LASV infection at steady state (i.e., are unchanging in time). This derivation, in turn, is used to translate the regression model's predictions of LASV seroprevalence into spillover infections per year in humans. For the model, we employ several assumptions: 1) humans within each 0.05x0.05˚ pixel constitute a closed population with constant per-capita death rate $d$. To facilitate steady-state analysis, we assume that new individuals are born in the pixel at a density-independent birth rate $b$. Within each pixel, humans are compartmentalized into three non-overlapping classes: susceptible ($S$), infected with LASV ($I$), and recovered from LASV infection ($R$). The size of the human population is assumed to be large enough so that stochastic events (LASV extinction) do not occur. 2) All LASV infections in humans are caused by contact with infectious rodents. Though human-to-human transmission of LASV is common in nosocomial outbreaks, rodent-to-human transmission is believed to be the primary pathway by which the virus is spread outside of hospital environments [15]. 3) Susceptible humans become infected with LASV at a constant rate $FS$, where $F$ denotes the rate of infectious contact between a human and infected *M. natalensis* (i.e., the force of infection). Any seasonal fluctuation in the contact rate between humans and rodents, as well as fluctuation in the prevalence of LASV infection in rodents, is assumed to average out over the decades-long timescales we consider. LASV-infected humans transition out of the infected class at per-capita rate $\gamma$; a fraction $\mu$ die from illness associated with Lassa Fever. 4) The remaining fraction $1 - \mu$ of individuals recover from infection and gain immunity from LASV.

The duration of LASV immunity in humans is not fully understood. Studies suggest that LASV immunity is the result of a combination of antibodies and a cell-mediated immune response [17, 67]. Anecdotal cases have shown that LASV IgG antibodies can remain in the blood of individuals for decades [10]. However, other studies have indicated that the level of LASV antibodies, as well as the extent to which an individual is protected against subsequent LASV infection, can wane with time [17, 67]. Preliminary analyses indicated that the possibility of waning immunity substantially influenced our model's estimates of LASV infections per year. Because of this uncertainty, we model the general scenario in which recovered

individuals lose immunity to LASV at per-capita rate $\lambda$ and transition back into the susceptible class. This more general model structure includes the scenario of lifelong immunity in the case that $\lambda = 0$.

Within each pixel across West Africa, the assumptions above lead to a system of equations that describes the number of humans in each of the classes:

$$\frac{dS}{dt} = b - dS - FS + \lambda R,$$

$$\frac{dI}{dt} = FS - dI - \gamma I, \tag{1}$$

$$\frac{dR}{dt} = \gamma(1 - \mu)I - dR - \lambda R.$$

We assume that, within each pixel, the dynamical system given by System (1) is at steady state. Consequently, the net rate of mortality is equal to the constant birth rate $b$, and each of the classes $S$, $I$, and $R$ are not changing with time. The corresponding steady-state values are found by setting the left-hand-side of Eq (1) to zero, and solving the resulting algebraic equations for each state variable. This yields the steady-state values

$$S^* = \frac{b(\gamma + d)(d + \lambda)}{d\lambda(\gamma + d + F) + d(\gamma + d)(d + F) + \gamma F\lambda\mu},$$

$$I^* = \frac{bF(d + \lambda)}{d\lambda(\gamma + d + F) + d(\gamma + d)(d + F) + \gamma F\lambda\mu}, \tag{2}$$

$$R^* = \frac{b\gamma F(1 - \mu)}{d\lambda(\gamma + d + F) + d(\gamma + d)(d + F) + \gamma F\lambda\mu}.$$

At steady state, the total population size in a pixel is $P^* = S^* + I^* + R^*$. We can write $P^*$ in terms of the model parameters by plugging in the steady-state values given by Eq (2):

$$P^* = b\frac{\gamma\lambda + d^2 + d(\gamma + F + \lambda) + F(\gamma + \lambda - \gamma\mu)}{d\lambda(\gamma + d + F) + d(\gamma + d)(d + F) + \gamma F\lambda\mu}. \tag{3}$$

By dividing $R^*$ by the total population size at steady state, $P^*$, we derive an equation for the steady-state seroprevalence, denoted $\Omega^*$:

$$\Omega^* = \frac{\gamma F(1 - \mu)}{\gamma\lambda + d^2 + d(\gamma + F + \lambda) + F(\gamma + \lambda - \gamma\mu)}. \tag{4}$$

Now we solve for the total LASV spillover rate $FS$, given that the steady-state LASV seroprevalence is $\Omega^*$. Solving Eq (4) for $F$ in terms of $\Omega^*$ yields:

$$F = -\frac{\Omega^*(\gamma + d)(d + \lambda)}{\Omega^*d + \gamma(-\Omega^*\mu + \Omega^* + \mu - 1) + \Omega^*\lambda}. \tag{5}$$

The rate of new infections is given by

$$\eta := FS^* = \frac{P^*\Omega^*(d + \gamma)(d + \lambda)}{\gamma(1 - \mu)}. \tag{6}$$

These analyses were derived using Mathematica. The notebook file is available in the github repository [23].

By substituting our prediction of human LASV seroprevalence for $\Omega^*$, we can estimate the total human infection rate using Eq (6). Calculating these estimates requires values for $d$, $\gamma$, $\mu$, $\lambda$, and $P^*$. We chose parameters that are broadly in line with the epidemiology of LASV and the demography of humans in West Africa.

We use values of death rate $d$ derived from country-specific lifespan estimates obtained from WorldBank [68]. For a pixel within a given country, $d$ is set to be the reciprocal mean lifespan of that country's 2018 life expectancy at birth. Studies indicate that the duration of LASV infection is typically about one month, so that $\gamma = 12 \ yr^{-1}$ across all pixels [11]. LASV infection causes mortality in a fraction $\mu = 0.02$ of non-nosocomial infections [17].

The rate of seroreversion is difficult to estimate empirically. McCormick et al. (1987) estimated that $\lambda = 0.064 \ yr^{-1}$ using a longitudinal study of IgG immune markers in individuals. However, it is unclear whether their results indicated true seroreversion, or whether the reduction of LASV immune markers below detectable levels made it appear as though seroreversion occurred. To better understand the potential consequences of seroreversion in our infection-rate estimates, we focus on two scenarios. In the first, any individual that has recovered from LASV infection remains seropositive for the remainder of their life ($\lambda = 0 \ yr^{-1}$). In the second scenario, seroreversion occurs at the rate estimated by McCormick et al. (1987) ($\lambda = 0.064 \ yr^{-1}$). In this case, an individual recovered from LASV is assumed to produce antibodies and maintain LASV immunity for an average duration of 15.6 years. We use the unprocessed WorldPop 2020 population data (described in S1 Appendix) as an estimate of the steady-state population size, $P^*$, within each pixel of the original $0.0083°$ resolution.

## Results

### LASV risk layers

The $D_M$ layer is constructed by averaging the predictions of 25 boosted classification tree models. Across all 25 bootstrap fits, the average out-of-bag AUC was 0.68, with a standard deviation of 0.05. This AUC indicates that the model has a modest ability to correctly discriminate pixels in which *M. natalensis* has been captured from background pixels, and is similar to out-of-bag AUC scores obtained in another study with a similar assessment criterion [5]. The algorithm assigned a high likelihood of occurrence to regions with a strong seasonal pattern of vegetation as well as specific levels of rainfall (S1 Appendix). Across 25 fitted models that made up the $D_L$ layer, the average AUC was 0.85, with a standard deviation of 0.08. This indicates a model that is good at discriminating between Lassa presences and absences. The algorithm primarily used precipitation contingency to determine whether or not a pixel is suitable for endemic LASV in *M. natalensis* (S1 Appendix).

Fig 3A–3C show maps of each of the fitted risk layers, as well as the combined layer of realized risk, $D_X$. As indicated by the IUCN range map for *M. natalensis* [19], most countries of West Africa are predicted to harbor this primary rodent reservoir of LASV (Fig 3A). However, the rodent is predicted to be less prevalent along coastal areas of West Africa and southern Nigeria. Similar to other Lassa risk maps [4, 5], our $D_L$ layer predictions indicate that the risk of LASV in rodents is primarily concentrated in the eastern and western extremes of West Africa (Fig 3B). The combined risk, shown in Fig 3C, indicates that environmental features suitable for rodent-to-human LASV transmission are primarily located in Sierra Leone, Guinea, and Nigeria.

### Connection to human seroprevalence and spillover

A quasi-binomial regression indicated a significant, positive association between the combined LASV risk predictor $D_X$, and the human arenavirus seroprevalence measured in serosurveys

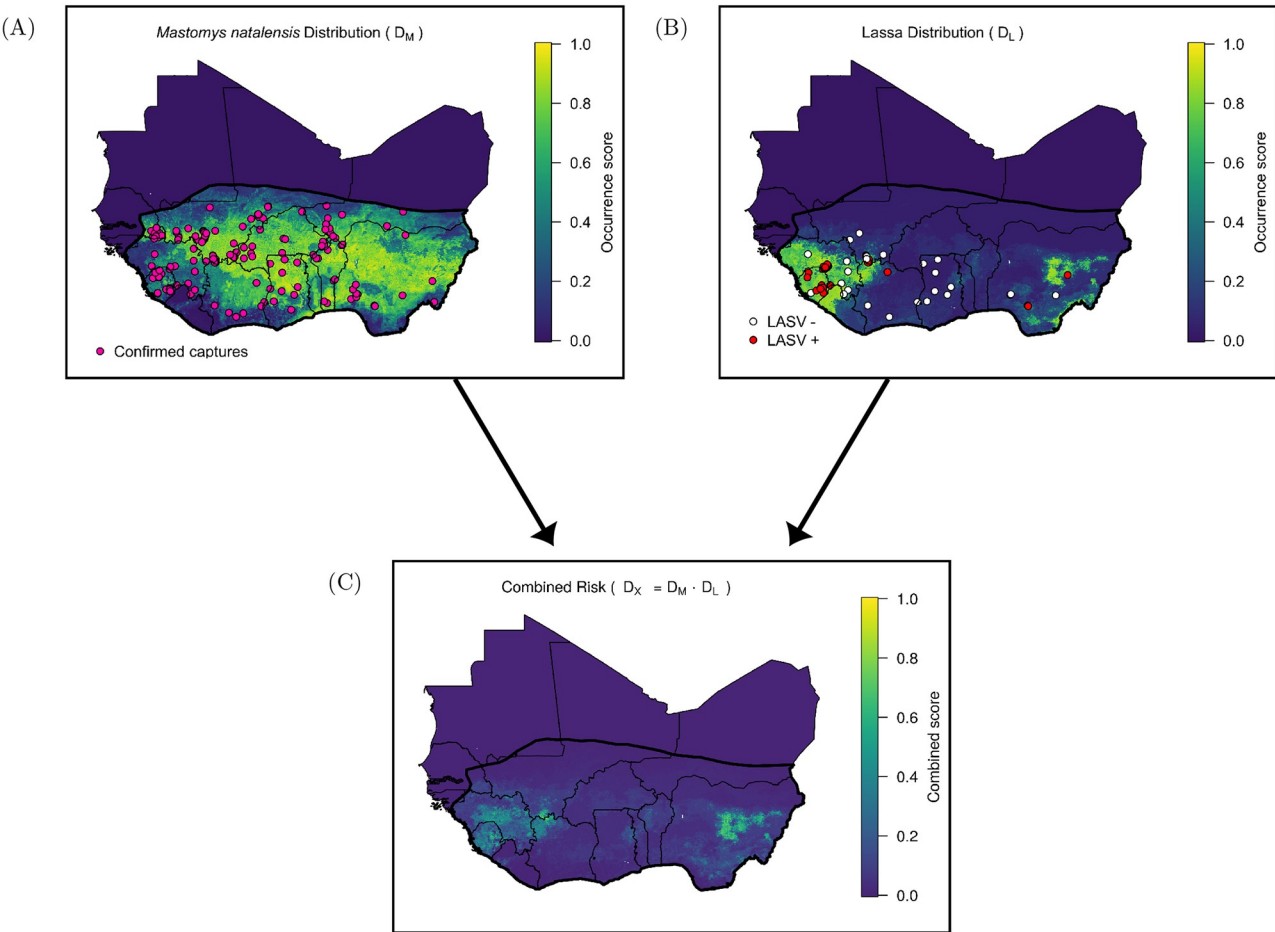

**Fig 3. Calculating the combined risk layer.** (A) Map shows the likelihood that each 0.05˚ pixel in West Africa contains the primary reservoir of Lassa virus, *M. natalensis*. Pink dots indicate locations of captures that were used to train the model. Black line indicates the IUCN *M. natalensis* range map. (B) Predicted distribution of Lassa virus in *M. natalensis*. Dots indicate locations in which *M. natalensis* were surveyed for the virus. (C) Combined risk, defined as the product of the above two layers.

(coefficient: 1.50, p = 0.000123, Fig 4). The model also indicated the presence of substantial overdispersion in the human seroprevalence dataset ($\varphi = 15.1$). More information on the GLM output can be found in the S1 Appendix. By applying the general linear model to the combined LASV risk layer, we extrapolate the human LASV seroprevalence across West Africa (Fig 5). Our results indicate that human LASV seroprevalence is greatest in the eastern and western regions of West Africa, with especially high seroprevalence in Central Guinea, Sierra Leone, and Nigeria.

Furthermore, by assuming that our predictions are representative of LASV infection at steady state, we can derive the number of LASV infections per year in humans. If the $D_X$ layer accurately describes the spatial heterogeneity of LASV seroprevalence in humans, and if LASV antibody production upon recovery is lifelong, our framework estimates that 897,700 new human infections occur each year. Between 664,300–843,800 (i.e., 74–94%) of these infections are expected to be sub-clinical or asymptomatic, leaving 53,900–233,400 infections that might require hospitalization [17]. Given that 2% of all infections result in fatality, our estimates imply that 18,000 individuals die of Lassa Fever in West Africa each year. Though our model does not account for differences of LASV risk by sex or age, research suggests that

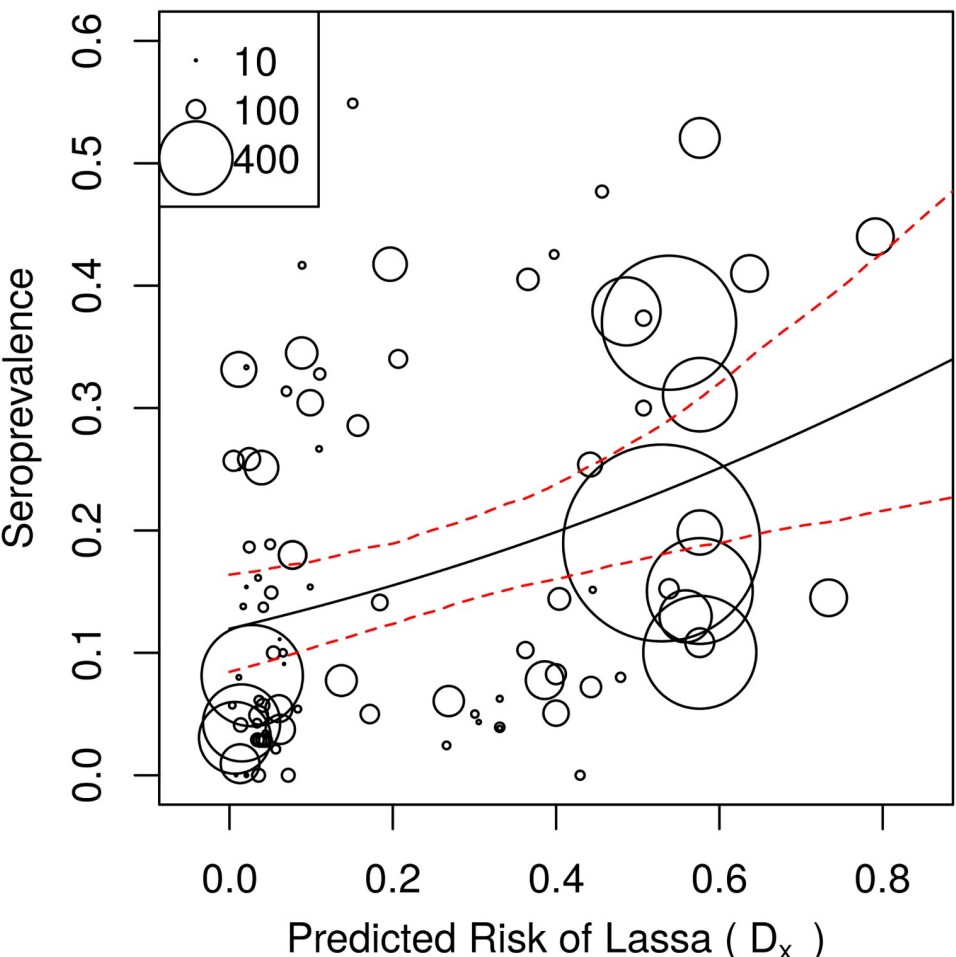

**Fig 4. Human arenavirus seroprevalence vs the combined risk layer.** Each circle represents a different serosurvey. The size of the circle indicates the number of humans that were tested. Solid black line shows the quasi-binomial prediction of seroprevalence, and the red dashed lines show the 95% confidence intervals. Confidence intervals were obtained by fitting the model 1000 times on random samples taken from the dataset with replacement.

hospitalizations may be skewed towards females, and fatalities will be biased towards individuals under 29 years of age but not skewed by gender [69].

Table 4 shows the number of LASV infections per year by country, ordered by number of infections. Our predictions indicate that more than half of new human LASV infections (531,700) in West Africa will occur in Nigeria (Fig 6). This distribution of LASV infection is largely due to the greater population size within Nigeria, as the per person spillover rates do not differ dramatically between countries (Table 4). After Nigeria, Ghana (60,200 infections per year) and the Ivory Coast (57,700 infections per year), respectively, are predicted to have the highest incidence of human LASV infections. Sierra Leone, Nigeria, and Guinea are predicted to have the highest per-capita rates of LASV infection (Table 4).

The above estimates are based on the premise that, upon recovery from LASV infection, an individual produces antibodies for the remainder of their life. If, instead, LASV antibody production ceases after an average of 15.6 years as suggested by some longitudinal serosurveys [17], then a given level of seroprevalence implies almost five times as many infections compared to the scenario with lifelong antibody production. Specifically, allowing for seroreversion

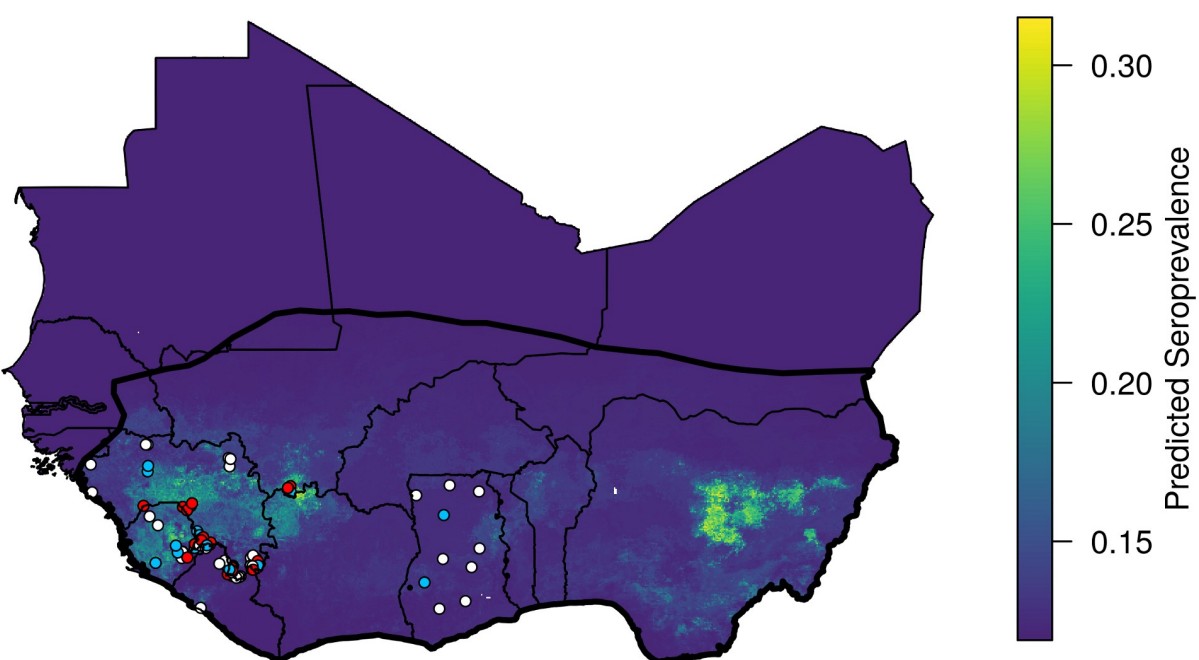

**Fig 5. Predicted human seroprevalence of Lassa virus in West Africa.** Dots show locations of human serosurveys, and dot color indicates the residual of the predicted seroprevalence. White dots indicate locations for which measured seroprevalence fell within 0.1 of the prediction. Measured seroprevalence at red dots was 0.1 or more greater than that predicted, and seroprevalence at blue dots was 0.1 or more below the prediction.

and subsequent LASV reinfection in the model implies 4,383,600 infections occur each year. Inclusion of reinfection does not change the ranking of countries in Table 4.

## Discussion

Machine learning approaches that forecast the spatial risk of emerging infectious diseases such as Lassa virus are often not designed to explain how aspects of the environment translate into

**Table 4. Predicted annual number of Lassa virus infections and infection rate.**

| Country | 1000's of infections | Rate |
|---|---|---|
| Nigeria | 531.7 | 2.6 |
| Ghana | 60.2 | 2.0 |
| Ivory Coast | 57.7 | 2.3 |
| Niger | 46.9 | 2.0 |
| Burkina Faso | 44.4 | 2.1 |
| Mali | 44.3 | 2.2 |
| Guinea | 35.0 | 2.5 |
| Benin | 27.0 | 2.2 |
| Sierra Leone | 20.7 | 2.9 |
| Togo | 17.9 | 2.2 |
| Liberia | 9.9 | 2.0 |
| Mauritania | 1.0 | 1.9 |
| Senegal | 0.8 | 2.0 |

Infection rate is in units of number of infections per year per 1000 people. Estimates in the table are derived assuming seroreversion and reinfection do not occur.

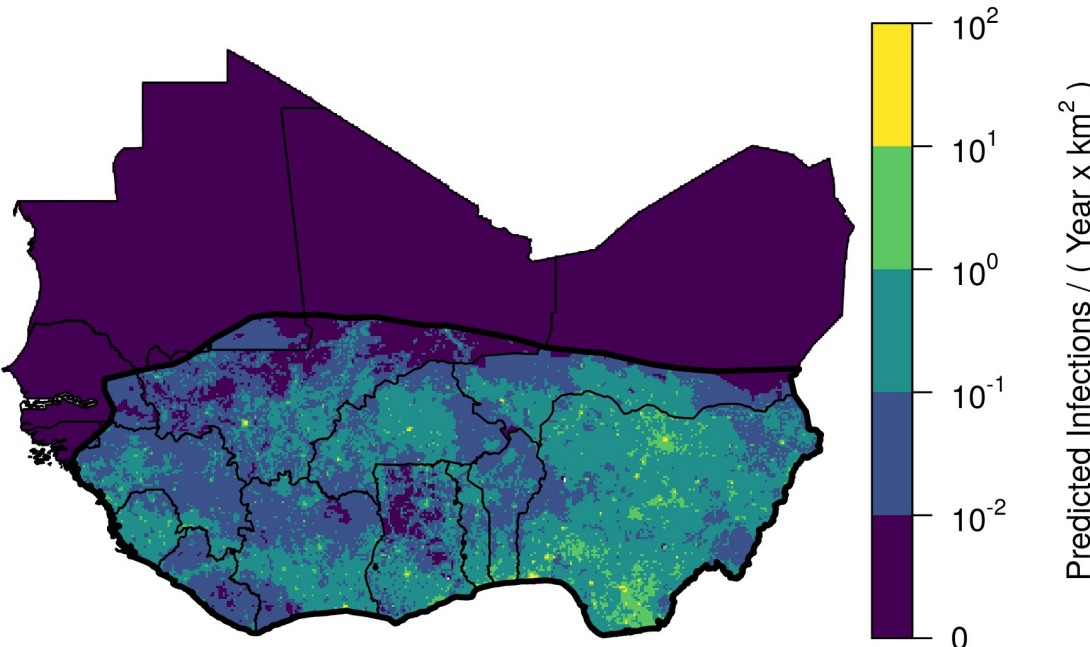

**Fig 6. Predicted spatial density of Lassa virus infections in humans.** Map shows the predicted infections per km². Yellow colors, representing a high number of infections, tend to occur in areas with high human population density and a high predicted seroprevalence.

realized pathogen spillover into human populations [4, 5]. Models that specifically predict attributes of the reservoir from the environment, and use these predictions to quantify spillover into humans, offer a more mechanistic understanding of the current and future spatial variation in human disease [70]. Our forecasting framework advances these approaches by generating predictions of spillover risk based only on data from the primary rodent reservoir of LASV, and rigorously assessing our risk predictions on realized human spillover as measured by human arenavirus serosurveys. As indicated by a generalized linear regression, our reservoir-based model of spillover risk explains a modest and statistically significant amount of the spatial variation in human arenavirus seroprevalence.

Using this framework, we are able to generate predictions of the number of new LASV infections within different regions of West Africa. Our results indicate that Nigeria contributes the greatest number of new human infections each year, and that the magnitude of new infections in Nigeria is driven primarily by its greater human population density, rather than an increased per-capita risk. An assumption that drives this result is the density-dependent form of spillover in the model (i.e., *FS*), in which rodent-human interactions increase with human population density. This form is appropriate if rodent interactions are well-mixed in the human population. For example, if increases in human density were reflected in an increased number of humans per dwelling, then the LASV risk posed by single rodent in a household would increase with human population size. If these assumptions are correct, Nigeria is likely to represent the greatest risk of LASV emergence because the large number of annual spillover events allows for extensive sampling of viral strain diversity and repeated opportunities for viral adaptation to the human population [71].

Our approach allows us to highlight the regions that contribute most to pathogen spillover, and suggest locations for further surveillance. Our model indicates that the highest per-capita risk to humans occurs in Sierra Leone, Guinea, and Nigeria. Given the data that are currently

available, our model suggests that these countries should be prioritized for surveillance of LASV emergence in rodents and at-risk human populations. Human serosurveys of the general population are notably lacking in Nigeria, but have the potential to clarify the true magnitude of LASV spillover in West Africa. Although it is known that certain broad regions of West Africa have a long history of LASV spillover (e.g., Sierra Leone, Guinea, Nigeria), relatively little is known about the prevalence of LASV in rodents or humans in other regions (e.g., Togo, Benin, Mali, Burkina Faso). Our model suggests that Lassa virus infections occur regularly in these under-sampled areas. Human serosurveys and rodent LASV testing from these regions could help modeling approaches clarify the spatial distribution of Lassa fever across West Africa.

In addition to identifying the regions most at risk for viral emergence, our model framework provides updated estimates for the rate of LASV spillover across West Africa. Previous estimates of 100,000–300,000 infections per year were based on longitudinal studies from communities in Sierra Leone conducted in the 1980s [17]. Using seroprevalence data from studies across West Africa, our model predicts between 897,700–4,383,600 LASV infections in humans occur each year. As demonstrated by past research focused on estimating LASV infection in humans, where the true value lies within this range depends on whether or not seroreversion and subsequent LASV reinfection are regular features of human LASV epidemiology, and therefore reinforces the need to better understand the scope for LASV reinfection [72]. It is important to realize that our predictions include both symptomatic and asymptomatic infections. Thus, because many human LASV infections result in mild flu-like symptoms or are asymptomatic, it is unsurprising that our predicted values exceed the reported number of confirmed LF cases in Nigeria [73, 74]. Several factors may contribute to the discrepancy between previous estimates of LASV spillover, and our revised estimates. McCormick et al. (1987) used seroconversion data from a 15 month period to infer a rate of LASV infection across West Africa. However, the population of West Africa has increased by a factor of 2.4 since that time, making these estimates outdated [75]. Furthermore, our estimates are based on human seroprevalence data that comes from five countries in West Africa and spans a 45 year time period. Because our dataset was obtained from a broader spatial and temporal range, our estimates are less likely to be biased by sporadic extremes in LASV spillover.

Accurate risk predictions could help guide risk-reduction and behavior-change communication campaigns, the distribution of future human LASV vaccines, and countermeasures directed at the rodent reservoir. In addition to vaccines that prevent infection in humans, new vaccine designs are currently being investigated for various wildlife pathogens as well, including pathogens in rodents [76, 77]. Wildlife vaccination campaigns that use vaccine baits have proven to be effective in the control of rabies in red fox (*Vulpes vulpes*) over large land areas, but require substantial planning and surveillance of the reservoir population [78]. Rodent population management could be another method of attenuating the risk of LASV in an area. Pinpointing areas that are most in need of spillover intervention will help overcome the logistical challenges that are associated with vaccine distribution to humans or wildlife on large scales. In addition to guiding intervention to specific regions, mechanistic forecasts similar to ours could help plan the logistics of such operations.

Our framework sheds light on the connection between LASV spillover in humans and the environmental conditions favorable to pathogen and reservoir. The reservoir layer of our model identified strong seasonal trends in vegetation (NDVI) as the primary explanatory variable that determines where the rodent *M. natalensis* occurs. This builds on other work that identified properties of vegetation as important predictors of the range of *M. natalensis* [5]. In conjunction with a strong seasonality of vegetation, our model identified a range of mean and

maximum rainfall values that limit the distribution of the LASV reservoir. This is in line with previous ecological studies showing that seasonal patterns of precipitation and vegetation are important drivers of seasonal breeding in *M. natalensis* [79]. Our model indicates that *M. natalensis* do not occur in areas associated with too much rainfall or areas without a clear wet/ dry seasonality, resulting in a lower risk of LASV spillover in coastal areas of West Africa and southern Nigeria. The pathogen layer of our model also indicates that strong seasonal precipitation patterns are the leading environmental feature that is associated with LASV in *M. natalensis* and the main driver of the LASV's occurrence in only western and eastern West Africa. Though the mechanism by which rainfall affects viral prevalence is unclear, it has been hypothesized, for example, that wetter conditions might facilitate the virus' ability to survive outside the host [4].

Our model of spillover risk predicts a significant, but small amount of the spatial variation in arenavirus seroprevalence studies in humans. The modest relationship between human LASV spillover and predicted risk might be due to the binary classifiers' coarse description of the magnitude of LASV risk. As more longitudinal data become available, these binary models can be upgraded with more nuanced models that predict the time-varying density of *M. natalensis* and the prevalence of LASV among the rodent population. Alternatively, the low correlation could indicate that other predictors like human factors have a large influence on LASV spillover. Geographic differences in housing, cultural practices, and diet likely influence the extent of LASV spillover but are not included in our model. For instance, the use of rodent-proof housing materials (e.g., concrete vs mud) and abstaining from rodent hunting and consumption are known to affect the extent to which LASV is able to transmit between rodents and humans [16, 80]. The residuals of seroprevalence predictions from our model could help guide understanding of which human factors mitigate or facilitate LASV spillover. If human factors like housing type can be readily identified from serosurvey locations within West Africa, they could be incorporated in the human stage of the model that connects spillover risk to human seroprevalence.

Geographic variation in LASV and its primary reservoir may also be responsible for the modest fit of our model. For instance, across West Africa LASV consists of several clades [22]. If certain clades are better at infecting humans, then our model will tend to underestimate the rate of human infections in regions where such highly-infectious clades occur. Similarly, the *M. natalensis* reservoir is also divided into multiple clades [20]. Different *M. natalensis* clades may differ in their contact rates with humans or in their suitability as reservoir, further reducing our model's ability to predict spillover into humans. Some evidence for this latter possibility comes from arenavirsues that preferentially infect certain clades of *M. natalensis* [21]. Because our study region only includes West Africa, it is likely that the *M. natalensis* occurrences that our model is trained on are only from the A-I clade [20]. However, our forecast should be interpreted with caution in eastern Nigeria, where the transition zone into the A-II clade occurs. Future work integrating these factors may help improve our understanding of the spatial variation in human seroprevalence that is due to the spatial patterning of LASV and reservoir clades.

Another factor that could influence our model fit is the possibility that rodent species other than *M. natalensis* serve as reservoirs or interact with the primary reservoir in ways that decrease or increase risk. Though *M. natalensis* is believed to be the primary reservoir that contributes to human infection, several species of rodents are known to be capable of harboring the virus [48]. Understanding the relationship between the habitat suitability of different rodent reservoirs and human LF burden may help determine whether *M. natalensis* is the host at which intervention strategies should always be directed. Furthermore, other species of

rodent may displace *M. natalensis* and therefore lower the overall spillover risk of LASV into humans. The layered framework we have developed can be easily adapted to include additional reservoir species and systematically investigate these possibilities.

Our model is constructed to learn and explain spatial variation in the average historical spillover of LASV, and does not include temporal trends of spillover risk. Due to the sparsity of available longitudinal data, our model assumes that the human population in West Africa, human LASV seroprevalence, and the rate of LASV spillover, are all constant in time. Over decades-long timescales, the rate of LASV spillover is likely increasing due to increasing rates of human-rodent interaction that come with urban growth, deforestation, or climate change [11, 70]. Estimating the combined temporal and spatial variation of infection will require long-term longitudinal studies in both rodents and humans across West Africa. With this data, for example, more advanced models could mechanistically associate an increasing rate of spillover with changes to land cover.

Another important temporal simplification of our current modeling work is the absence of seasonality in LASV spillover. In Sierra Leone, Guinea, and Nigeria, hospital admissions attributable to LASV infection generally peak late in the dry season [54, 69, 81]. In these regions, the mechanism of seasonal spillover likely involves a combination of seasonal rainfall and land use practices, such as crop-harvesting and subsequent burning of agricultural fields, that drive rodents into domestic dwellings in search of food-stuffs [54, 82]. It is not understood whether these factors operate uniformly across all of West Africa. Temporal fluctuations in the density of the reservoir population, due to seasonal cycles of reproduction, are another potentially important factor that could drive a seasonal spike of human LF cases. However, it is unclear whether the density fluctuations that have been observed outside of the LASV geographic range (e.g., Tanzania [79]) also occur within West Africa. At least in Guinea and Sierra Leone, research on the population dynamics of *M. natalensis* indicates that density fluctuations are much weaker than those in East Africa [83]. In the case of rodent vaccination, understanding population dynamics is particularly important because distributing vaccines at seasonal population lows in wildlife demographic cycles can, in theory, substantially increase the probability of pathogen elimination [83, 84].

Although the methods we have used here make efficient use of available data, the accuracy of our risk forecasts remains difficult to rigorously evaluate due to the limited availability of current data from human populations across West Africa. The sparseness of modern human data arises for two reasons: 1) the lack of robust surveillance and testing across much of the region where LASV is endemic and 2) the absence of publicly available databases reporting human cases in those countries that do have relatively robust surveillance in place (i.e., Nigeria). Improving surveillance for LASV across West Africa and developing publicly available resources for sharing the resulting data would allow more robust risk predictions to be developed and facilitate risk reducing interventions. Despite these limitations of existing data, the structured machine-learning models we develop here provide insight into what aspects of environment, reservoir, and virus, contribute to spillover, and the potential risk of subsequent emergence into the human population. By understanding these connections, we can design and deploy more effective intervention and surveillance strategies that work in tandem to reduce disease burden and enhance global health security.

## Supporting information

**S1 Appendix. Details on the predictors used in the model and model fits.**
(PDF)

## Author Contributions

**Conceptualization:** Andrew J. Basinski, Christopher H. Remien, Scott L. Nuismer.

**Data curation:** Andrew J. Basinski, Elisabeth Fichet-Calvet, Anna R. Sjodin, Tanner J. Varrelman.

**Formal analysis:** Andrew J. Basinski.

**Funding acquisition:** Brian H. Bird, Peter A. Barry, Michael A. Jarvis, Scott L. Nuismer.

**Methodology:** Scott L. Nuismer.

**Software:** Andrew J. Basinski.

**Validation:** Brian H. Bird.

**Writing – original draft:** Andrew J. Basinski, Scott L. Nuismer.

**Writing – review & editing:** Andrew J. Basinski, Anna R. Sjodin, Christopher H. Remien, Nathan C. Layman, Brian H. Bird, David J. Wolking, Corina Monagin, Bruno M. Ghersi, Peter A. Barry, Michael A. Jarvis, Paul E. Gessler.

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
