## [Decision Letter · Decision Letter 0]

27 Sep 2020

Dear Dr. Basinski,

Thank you very much for submitting your manuscript "Bridging the gap: Using reservoir ecology and human serosurveys to estimate Lassa virus incidence in West Africa" for consideration at PLOS Computational Biology.

As with all papers reviewed by the journal, your manuscript was reviewed by members of the editorial board and by several independent reviewers. In light of the reviews (below this email), we would like to invite the resubmission of a significantly-revised version that takes into account the reviewers' comments. A number of the reviewers have raised a number of very important and justified concerns that would be need to be addressed. 

We cannot make any decision about publication until we have seen the revised manuscript and your response to the reviewers' comments. Your revised manuscript is also likely to be sent to reviewers for further evaluation.

Sincerely,

Amy Wesolowski

Associate Editor

PLOS Computational Biology

Nina Fefferman

Deputy Editor

PLOS Computational Biology

Reviewer's Responses to Questions

**Comments to the Authors:**

Reviewer #1: In this paper, the authors take a machine learning approach to predicting rates of Lassa virus (LASV) spillover from its reservoir host, Mastomys natalensis, to human populations in West Africa. The authors use an approach that is distinct from previous studies by training their model exclusively on the reservoir data to derive a metric of regional spillover risk from (a) presence/absence studies of the reservoir rodent species, Mastomys natalensis, combined with (b) field studies testing for the presence/absence of LASV infection in M. natalensis in the same locality. The authors then use the pixel-by-pixel output layer from these combined predictions to derive a measure of the force of infection that drives zoonotic spillover of rodent-borne LASV to the human population. They then solve an SIRS model at endemic equilibrium for human populations at each corresponding locality, assuming infection to be due to spillover alone and compare the estimated “Recovered” population proportion at endemic equilibrium to previous measures of human seroprevalence in the region. The authors find a weak correlation between their predictions for human steady state seroprevalence by region and the actual data.

The paper does an admirable job attempting to actually test some of these machine learning predictions about “spillover risk” with actual data in the human population—a practice that has not been previously attempted. However, I am not convinced that their test measure (seroprevalence in the human population) has any real biological significance, and I am concerned about the steady state assumptions of the ODE, as well as the extremely high case predictions generated under assumptions of waning immunity when much of the LASV literature (e.g. Bausch et al 2013) suggests that human immunity to LASV infection is longterm. My main concerns can be summarized as follows:

1. On line 436 of the Discussion, the authors raise the issue of contrasting dynamics of multiple genotypes of LASV. This is a critically important consideration for LASV and also deserves consideration from the perspective of the reservoir host, since previous work has shown that that different sub-species of M. natalensis carry genetically distinct different arenaviruses all closely related to LASV but with varying capacity for infecting humans (see Gryseels et al. 2017 Plos Pathogens). I would imagine that human serological responses to a number of these arenaviruses may be impossible to differentiate, but modeling their dynamics collectively amongst a variety of hosts is not correct. At the very least, the authors need to include some discussion of these considerations. Better, I would like to see the sub-species of M. natalensis reported or at least summarized so that we can better understand the accuracy of these risk layers that treat all M. natalensis as equal.

2. The authors highlight that their spatial ‘risk’ model is likely to be more accurate than previous because it is limited to reservoir data only, and rodents are much more restricted in their movements than humans. That obviously could explain why their risk predictions only show a weak association with regional human seroprevalence (e.g. the humans are moving around), but it is important to note too that there is vast uncertainty in the regions where essentially no surveillance has taken place, as rodents are likely to be sampled (and in particular tested for LASV) in regions where humans are known to have been previously infected. In particular, the authors include Togo and Benin in their risk map, but it appears that neither country has reported any serosurveillance of rodents for LASV, which does not mean that these regions are low risk but rather that we don’t know anything about the.

3. The second layer pixel map for whether a region is deemed to be LASV positive vs. negative is derived from PCR tests of M. natalensis captures for LASV. Intriguingly, the authors choose to represent the region as LASV negative if either serology or PCR-based results in rodents show no evidence of LASV infection. This is reasonable, given that a seronegative result shows no back history of exposure. However, the authors only designate a region as LASV positive if there is PCR-based detection of LASV infection in M. natalensis, and no mention is made of what is done if there is not PCR-based evidence but if the rodent population showed seropositivity. I am concerned that PCR-based detection is unlikely to pick up on seasonal shedding of virus (which we know is important for LASV – see Akhmetzhanov et al. 2019 Phil Trans Roy Soc), but the metadata posted to github appear to not show the seasonality of infection. In fact, some of the diagnostic methods in the data are listed as “various” which provides no information at all. I would like to understand if there were positive serological data in the rodents that were excluded and why.

4. Point #3 leads me to also highlight the absence of incorporation of any signature of seasonality in infection risk in this paper – risk is measured exclusively as a spatial effect under equilibrium assumptions which just is not accurate. We know full well that there is a risk season for Lassa which takes place in the Dec-Jan months that correspond to the dry season and the rodent breeding season and this is barely even discussed, let alone explored in detail.

5. In addition, building on point 4, the authors use a somewhat strange approach to predicting seroprevalence spatially, assuming the absence of spatial fadeouts of human infection which are almost a certainty for a rare zoonosis with a distinctive seasonality. It would be much more compelling if they could make this a catalytic model predicting age-seroprevalence (e.g. Muench 1959) and a constant hazard of seroconverting across an individual’s lifespan. I did not go through all the reported human data in detail, but at least a subset of it appears to have age data associated. This paper would be much stronger if the authors made an attempt at this.

6. Additionally, the authors report estimated seroprevalence and incidence for assumptions of both SIRS and SIR. There are certainly compelling studies out there suggesting that human Ab immunity to LASV can be lifelong (e.g. Bausch et al. 2013 ASTMH) which contradicts the SIRS assumptions and provides a more reasonable (i.e. lower) estimate of cases. I would like to at least see the emphasis on SIRS justified and it would be useful to see some comparison of a boosting model where low Ab titers wane but can be boosted upon re-exposure. Given that humans are not modeled as infectious, it should not change the dynamics but it might fit the data better… Which leads me to ask—how well do these models fit the data? I would love to see some statistical output of that line’s fit to the data in Figure 4.

A few additional line-by-line comments are listed below:

Author Summary:

- SARS-2 coronavirus is incorrect terminology. It is SARS-CoV-2 and coronavirus is already represented in the middle of the term

Main text

- Line 44: It is important to caution that these assumptions probably don’t hold for all reservoir species (e.g. bats are highly mobile)

- Line 49: I know that ambisense viruses are still classed within the negative sense viruses for taxonomic convenience but it is a bit strange here to see that you class Lassa as both negative sense and ambisense. Suggestion to just stick to ambisense here.

- Line 91-92: What about sub-species? This is super important to Lassa virus dynamics – see comments above

- Line 114: PCR tests really need a seasonal component to pull off this Lassa positive/negative risk map. Why are serological studies not included here but are included in evaluation of Lassa negative regions? See comments above.

- Line 254: Spillover risk is modeled as FS, where F is the regional FOI from the reservoir and S is the human population density, which assumes this to be a density-dependent transmission process. Can you justify this assumption? This is what leads you to conclude that Nigeria is at the highest risk (they have the most people), but that does not necessarily mean that there will be a higher number of human-rodent contacts. Unless rodent density increases with human density (which it may), it seems like a single rodent would have a fairly limited movement range and likely a discrete number of likely human contacts.

- Line 258: This lack of mortality effect is a strong assumption in a disease that has known mortality. I accept the need to model it this way, but how might consideration of this change results?

- Figure 1: It seems that you can’t say anything really about the risk of LASV spillover in Mauritania, Burkina Faso, Senegal, Niger, Benin, or Togo since there are no rodent infection data to actually evaluate.

- Figure 3: As in above, this figure highlights that the “risk map” of potential spillover for Lassa is really just an absence of surveillance. The regions where infected Mastomys have been previously tested pop up and those regions where they have not do not.

- Figure 4: Can you provide some statistical output on the findings in this figure?

- Line 345: This is extrapolating way too far – I would advocate for restrict the projections to the more conservative spatial estimates. It seems like this is reported here just to have a dramatic number.

- Great job with the SI and the github repo. Code and data are nicely organized and easily accessible.

Reviewer #2: Note: review comments also uploaded as a PDF, which is formatted better so may be preferable for the authors to work through.

Reviewer summary

This paper presents a method to predict zoonotic spillover risk from environmental data. Boosted classification trees were used to model the relationship between environmental variables, the reservoir population and animal Lassa virus prevalence to ultimately predict the incidence of human Lassa virus infection. The authors propose that the methodological novelty lies in the distinction between the vector distribution and pathogen reservoir layers (used to generate predictions) and the human seroprevalence layer (used to model the link between predictions and observations). The model predicts a far greater number of Lassa virus infections than current, outdated estimates and predicts their distribution across countries in sub-Saharan Africa. These results imply that existing surveillance likely misses the majority of symptomatic and asymptomatic infections and highlights key geographical areas for enhanced virus surveillance. Much of the discussion and implications pertain to future work and routes to understand the (often substantial) remaining variance between predicted and observed human seroprevalence. To me, the main hypothesis being tested is “does the presence of Lassa infected M. natalensis predict magnitude of human seroprevalence?”, and the results suggest “not very well on its own”.

I am not an expert on zoonosis nor Lassa fever, so I am reviewing this from the perspective of an interested infectious disease epidemiologist with a computational background. I apologize if my questions or comments on the disease natural history seem obvious or silly!

The paper is very well written; the flow between sections is great, the introduction sets the scene nicely, and I left the methods feeling qualified to understand the remaining sections. The code sharing and accompanying readme are clear, and I was able to re-run their analyses with no problems (though I do not have a Mathematica license).

This paper appears to be an important next step in a fairly sparse literature (at least from my cursory searching). However, right now it is predominantly an eloquent presentation of a method (with some novel ideas) but with only a cursory consideration for the biological implications. I am also a little confused by the methods: combining the two layers by multiplying them, then using the binary classifier score as a predictor for a continuous outcome in the GLM seems like a weird pipeline. Though this may be my misunderstanding. Given that the binary classification score (from the first two layers) appears to have little explanatory power in the GLM, I am not sure what the results add over the referenced Mylne et al. paper. I think these methodological concerns need to be clarified and the implications of the results (rather than just the unrealized potential of the method) made clearer if this is to warrant publication in PLoS Comp Biol.

Major comments

- The authors claim to be modelling spillover risk, but I am not convinced that is what their model is predicting. I am probably missing something crucial, but it seems odd to train the model on binary data (presence/absence of LASV or M. natalensis) and to then use the predictions as a quantitative estimate of spillover risk. The model may predict a high classification score for presence of LASV infected rodents in a pixel, but does that necessarily mean that it is predicting higher LASV prevalence in the reservoir population? Is it distinguishing between a small rodent population with low prevalence and a large population with high prevalence? It seems like those data points are weighted the same here. Therefore, from what I understand, the method is only predicting the binary presence or absence of at least 1 LASV infected rodent which is what Mylne et al. did (they call it suitability index rather than spillover risk). If so, what do the authors think the GLM would look like if Mylne et al’s suitability index were used instead?

- The GLM clearly does work given that it explains some variance and the slope is significant. However, if the predictor is only binary, then isn’t the comparison to magnitude of human seroprevalence somewhat flawed? For example, a classifier for “absence, low LASV prevalence in rodents, high prevalence LASV prevalence in rodents” would make more sense if the aim is to predict magnitude of risk. Is a lot of quantitative information being thrown away converting to a binary prediction, and is this why the GLM appears to explain such little variance? Maybe I am trying to be too mechanistic, but I think this would be helped if the methods and results clarified why this approach makes sense.

- Figure 2 and associated methods: I have been trying to intuit why simply multiplying the likelihoods makes sense, but something feels off. The D_M layer is predicting the likelihood of M. natalensis presence given the environmental variables, whereas the D_L layer is predicted the likelihood of LASV circulation given the environmental variables and the presence of M. natalensis. I would recommend adding some text to explain the intuition behind this. A bit of a half-baked thought, is it related to the conditional probability P(LASV | rodents) * P(rodents)? If so, is multiplying the likelihoods the right approach?

- Discussion of the implications: does this new denominator change our understanding of the Lassa fever clinical spectrum and epidemiology? For example, if the IFR is 2%, then 3 million infections implies 60,000 deaths. Is this consistent with underreporting of LF-associated deaths, or does it imply that the IFR is set too high?

- A positive comment – combining a SIRS model with predicted seroprevalence and immune waning estimates to get incidence is really neat!

- Figure 4 is pretty underwhelming: it leaves the reader feeling unconvinced that the modelled relationship is real or useful. The plot should at least have confidence intervals on the model line. Also is this from the weighted or unweighted version? And relating to Figure 5 (see minor comments), is this actually doing much better than assigning each pixel the mean seroprevalence? The authors derive a null model for Lassa seroprevalence but don’t really use it in this way.

- The discussion is well written and raises a lot of interesting points. However, it does not discuss the results from the D_M and D_L layers at all. For example, it is not clear how to interpret the result that maximum precipitation and precipitation contingency are the 2 most important predictors in both layers. Are the key environmental predictors the same or different to previous work, and are there any novel biological insights there? L467 hints at this, but I didn’t feel like the discussion clarified what these insights are. There is some text in the supplement, so this should at least be referred to.

Minor comments

- The omission of a time component in the model seems important. The reader doesn’t know when e.g., the rodents were captured relative to e.g., when seroprevalence was measured. Is there some evidence to support the assumption of steady-state seroprevalence rather than increasing seroprevalence over time? Are there any longitudinal or repeated cross-sectional studies?

- I think there is an opportunity to further test the sensitivity of the D_M layer. You have an independent dataset for true positive M. natalensis presence: the data used to train the D_L layer. Does the D_M layer accurately predict the presence of M. natalensis in those locations? This might be a useful small supplementary analysis to support the accuracy of the D_M layer.

- The AUC for the same predictors (absence/presence of M. natalensis or Lassa) looks almost identical to those in Table 2 of Mylne et al. Does this mean anything, given that the top environmental predictors are the different?

- L41: It isn’t clear to me how the issue of human seroprevalence not coinciding spatially with location of infection is resolved here. When comparing the model predictions to observed seroprevalence in the GLM, is this issue not being re-introduced anyway? Consider clarifying.

- L70+ The date range for the animal and LASV data is not mentioned, but it is for the human seroprevalence data. Would be good to see this to understand how comparable the dates are.

- L70+: Similarly, the survey data are not described in much detail. I understand that there are so many sources that this would take up too much text, but a supplementary table, particularly for the human seroprevalence data, describing sample sizes, demography, protocol etc might be useful.

- L121: The text suggests that the human seroprevalence data are surveys of all arenaviruses antibodies, not specifically Lassa. Perhaps this is obvious to someone with experience with these pathogens, but is there a reason why the included studies were not LASV specific? Are the assays highly cross-reactive or are there no other arenaviruses that infect humans? This should be clarified.

- L130: I understand the decision to only include land types that have been consistent for 20 years from a methodological perspective, but does this not ignore a major source of spillover risk – increased activity at the animal-human interface? It’s not my area so I may be wrong, but if this is important, its implications and limitations should be discussed in the main text. From the abstract of the cited Gibb et al. paper: “Although the recent increase in LF case reports is likely due to improved surveillance, recent studies suggest that future socio-ecological changes in West Africa may drive increases in LF burden”.

- Related, L444 mentions time but not in the context of land type. Perhaps mention this here.

- L177: Is a Bonferroni correction for multiple comparisons appropriate here? Probably not given that this is just an arbitrary pre-filtering step, but something to consider.

- L207: Maybe I’m being slow, but I don’t follow the logic for how false negatives make the classifier conservative. Please clarify.

- L209: Dumb question, but is this spatial distance or minimum distance across environmental variables?

- L258: Is it not a fairly easy addition to the model to assume some deaths? 2% mortality is a lot of deaths when you’re predicting millions of infections! How would this relate to how we understand LF epidemiology now? Though I understand if this would mess up eq 2-4.

- L266: is there evidence that arenavirus seroprevalence is at steady state or is this an assumption?

- L280: Refers to the data section, but there is no information on the WorldPop data there. Is this text missing or is the reference erroneous? (I think it should refer to S1 Appendix).

- L282: Assumed a mean lifespan of 50 years for all locations. Why not use country specific estimates?

- L281: It took me an embarrassingly long time to intuit these equations. The equations are correct, but I couldn’t get my head around the idea of there being a constant “steady-state population size” if the real population size is growing (ie. when b > dN). I think I see now that this is a possible equilibrium because i) the incidence rate does not depend on the I compartment (as in a textbook example) and ii) crucially, the birth rate b is assumed to be not density dependent. I couldn’t understand how the incidence of new infections could be constant while the population is growing and the proportion in each compartment stays the same. But I think it works because b just feeds in a constant number of new susceptibles (rather than an increasing number if the term were bN, which I think would be more standard). Nothing to be done if this is a trivial realization, but some text to explain the intuition for why this steady state exists and the rational for a flat birth rate might help a confused and pedantic reader like me!

o I found page 16 of this: https://arxiv.org/pdf/2004.04675.pdf and this paper which might be helpful: https://doi.org/10.3390/math5010007. Not suggesting to cite.

o Also page 88 of the “an introduction to infectious disease modelling” by Vynnycky and White was useful.

- L292: At the moment, the authors present the immune waning estimate from McCormick et al. 1987 as an upper bound. Is there any evidence to suggest that waning might be even faster than the point estimate from McCormick 1987 (e.g., were there confidence intervals on lambda in that paper)? If so, this would be a useful scenario to present.

- L319: missing space before the bracket.

- Figure 5: a histogram of residuals might be a useful subplot here. It’s hard to assess how well the model is doing from this plot.

- L376: This is a really key paragraph, but a bit more information on the surveys, for example, predominant housing type in particular outliers above/below the prediction line, could be useful. Was this information not available to include in the GLM here?

- I don’t know this literature, but it looks like there may be a few relevant studies tackling a similar problem that are not discussed, e.g., https://besjournals.onlinelibrary.wiley.com/doi/full/10.1111/2041-210X.12549 which also provides quantitative spillover estimates

Comments on code

Really nicely documented, well written and clean code. Great job. I ran everything line by line and found no obvious bugs! Only small comment is to not use reserved words (eg. data, grid) for user defined objects.

Generate_Reservoir_Layer.r

L79: in the object `classi.dat.rod`, none of the pseudo-absence variables have countries or confidence. Just checking that this doesn’t impact anything downstream.

Reviewer #3: Improved methods to track and investigate determinants of zoonotic spillovers will enhance our ability to mitigate their risk to public health. I appreciate the authors’ efforts to improve the methods we use to understand these phenomenon, and to unravel remaining questions about the distribution of risk for human Lassa fever infections. Their inclusion of serosurveys into estimates is very reasonable and the approach could be of interest to readers.

The paper would benefit from a framing that includes more about what we do and do not know about the distribution of human risk for Lassa, and what we would do differently if we did know these things to motivate the study. Currently, the introduction includes elements from the methods, results, and discussion and the paper would be easier to follow if the role of each section of the paper were more clearly delineated.

More specific comments on each section are below:

Abstract: The abstract is very slim on results and offers few conclusions about what the next steps should be, given the new findings from this study. Consider cutting back on the background and rationale to make room for results and some discussion.

Introduction

Lines 3-4: Not all zoonotic pathogens circulate in a wildlife reservoir.

Lines 28-29: “As a result, the extent to which predicted risk explains the realized variation in human exposure to the pathogen is unclear.” Might one way to evaluate this also be surveys to better understand exposures?

The introduction would be more compelling if it focused on what is and is not known about risk to humans, a summary of what we already know about the frequency and distribution of human infections, and the gaps in our understanding of risk. Limits of the published models to estimate this risk are important to mention, but should also be placed within the context of measured seroprevalence. A clear statement of the study objective(s) would also be helpful for the reader.

Data

It would be useful to say something about when data were collected and the geographical extent they represent. It becomes clear as the paper goes along that the data are obtained through literature review, but it would be useful to have this stated upfront, perhaps with some of the methods used to identify the papers with data applicable to the analysis. I’m not clear on why the description of where the data come from is not included in the section called methods.

The secondary data used in this study span many decades – it would be useful for the reader to have a table outlining the location and year(s) for the data used in the models.

Line 94: The study area is unclear.

How are pixels defined?

Lines 116-118: Can the authors provide some rationale for the definition of positive and negative pixels? Are there any limitations to the approach authors used to define positive and negative pixels?

Lines 122-123: Regarding the requirement for individuals be sampled randomly, would it not be sufficient for households within a village to be sampled at random, and individuals from those households included? Additional details about the literature reviews, including the papers that were excluded from the study would be useful for the reader to understand the process. (See comment about summary table above, which could also possibly include details about sampling methods.)

Authors have restricted the serosurveys they included in the models, for good reason, to those that were conducted in some kind of random sample of the population. However, would be good to compare the known occurrences of infections with the resulting model predictions to see how they correspond.

What are the limitations of serosurveys in humans? For example, how good are the assays that measure previous infection and how long do antibodies last? Some information about the antibody response would be useful to inform interpretations of the yearly incidence estimates. If the authors’ primary objective is to estimate the number of infections, I would assume that limitations in the measurement of past infection and assumptions about interpretations of these results should be discussed in detail – both in the methods and discussion section. Since the resulting estimates of the numbers of cases each year from this analysis have a very wide range, might it be useful to conduct some sensitivity analyses around some of the key assumptions about what a serologic response means? The authors correctly note that greater insight into the natural history of infection would be useful, but which points in particular would be most important to understand better for this kind of model?

Results

Figure 5: There is little variation in the predicted seroprevalence across West Africa – although the ‘hotspots’ have a predicted 18-20% seroprevalence, 12% prevalence in other areas also seems high. Most of the serosurveys used to inform this model have estimates that are outside of the entire predicted range.

Lines 340-341: It’s unclear why the authors calculate estimated yearly infections using the as assumption of uniform risk.

Lines 345-346: What assumptions were made about duration of antibody protection to estimate reinfections? Additionally, information about the natural history of Lassa infections in humans would also be useful. How many infections are expected to be symptomatic? What proportion of people infected will die?

Are there age or gender differences in risk or in seroprevalence? How were these accounted for in the model?

Predicted seroprevalence is 12% even in areas without the known reservoir host. This seems like a limitation in the ability of the model to predict human risk. Indeed, the fit of the model isn’t great – would be good to include more about this in the discussion section.

Discussion

This section could be improved by reducing the overlap with the results.

Lines 383-385: Demographic and Health Surveys are routinely collected from West Africa and these include home building materials. These data are publicly available and could be incorporated into the model, as the authors suggest.

Lines 394-396: While the authors identified areas of relatively higher risk, or higher numbers of infections because of larger population sizes, it’s unclear why other areas that still have significant risk shouldn’t be targeted for surveillance and risk mitigation strategies. Why are relative differences in risk more important than absolute risk?

I would be interested in hearing more from the authors about the modest correlations with the human seroprevalence data. Indeed, many of those studies are very old and it is likely that the conditions that drive Lassa transmission have changed in the intervening years. What are the pros and cons of the data used for the models and why is the correlation so modest? What else are we missing about this disease system?

Lines 392-394: The authors mention multiple times the use of vaccines for rodents. As far as I know, there are no rodent vaccines for Lassa. There are efforts underway to develop human vaccines. It seems that part of the rationale for this study is to inform prevention; if true, it would be useful to provide a more informed discussion of the possible strategies and how their deployment might be facilitated by this research.

Line 444: The authors bring up the idea of Lassa elimination which seems to be an unrealistic goal for a pathogen endemic in a wildlife host. Are there examples of elimination from other disease systems that the authors can reference to support this idea?

The authors state (467-469): “…the structured machine-learning models we develop here provide insight into what aspects of environment, reservoir, and virus, contribute to spillover, and the potential risk of subsequent emergence into the human population.” It would be good to more completely discuss each of these points in terms of what was previously known and what we learned new from this study for each of these points to highlight the added value of this work.

**Have all data underlying the figures and results presented in the manuscript been provided?**

Reviewer #1: Yes

Reviewer #2: Yes

Reviewer #3: Yes

PLOS authors have the option to publish the peer review history of their article (what does this mean?). If published, this will include your full peer review and any attached files.

Reviewer #1: No

Reviewer #2: **Yes: **James Alexander Hay

Reviewer #3: No
---

## [Decision Letter · Decision Letter 1]

31 Dec 2020

Dear Dr. Basinski,

Thank you very much for submitting your manuscript "Bridging the gap: Using reservoir ecology and human serosurveys to estimate Lassa virus incidence in West Africa" for consideration at PLOS Computational Biology.

As with all papers reviewed by the journal, your manuscript was reviewed by members of the editorial board and by several independent reviewers. In light of the reviews (below this email), we still have concerns that key points in the requested resubmission were adequately addressed. In particular, there are three key points: 1) addressing how the very staggering number of estimated LASV human infections per year can be validated/justified to be realistic, 2) a human infection model using an age-seroprevalence model and 3) the use of SIRS dynamics over SIR. If these points can be addressed adequately, we would like to invite the resubmission of a significantly-revised version that takes into account the reviewers' comments. 

We cannot make any decision about publication until we have seen the revised manuscript and your response to the reviewers' comments. Your revised manuscript is also likely to be sent to reviewers for further evaluation.

Sincerely,

Amy Wesolowski

Associate Editor

PLOS Computational Biology

Nina Fefferman

Deputy Editor

PLOS Computational Biology

Reviewer's Responses to Questions

**Comments to the Authors:**

Reviewer #1: I reviewed this paper the first time some months back and raised the following main concerns:

1. Inability of the authors to distinguish data from different genotypes of LASV and/or sub-species of M. natalensis

2. Vast uncertainty in the accuracy of predictions from regions that have never been sampled for rodents, in particular Mastomys, at all

3. Confusion over whether sites that were LASV Ab+ for rodents but PCR- were considered LASV+ in the pathogen layer

4. Lack of seasonality included in either the reservoir or pathogen layer.

5. A desire to see human infections modeled using an age-seroprevalence approach.

6. Concern over the emphasis on SIRS dynamics over SIR with very little support for this decision.

On the whole, the authors have done a fairly thorough job addressing these concerns: they have directly adapted their analyses in response to points #1-3, discussed but not addressed #4 and 5, and taken a haphazard approach to their response to point #6. I can accept that #5 (age-seroprevalence) will not be explored in this paper but I still have questions over the two outstanding points:

1.Lack of seasonality included in either the reservoir or pathogen layer:

Point taken that dynamics are assessed over long timescales and that spatial, environmental factors are accounted for. However, given the sparseness of the data for some of these localities, as well as the short lifespan of the rodent hosts (meaning Ab+ data may be difficult to acquire), I still think it is possible that a few rodents are sampled at the wrong time of year to get a false negative LASV pixel result for a given region. Along these lines, do you really think 5 Ab negative rodents is enough to conclude that a site is LASV(-)? My understanding is that the average lifespan of M. natalensis is only ~6 months, so I would not be surprised at all to find 5 seronegative individuals in a site where Lassa really does occur. Can you provide a baseline seroprevalence in rodents of a LASV(+) site for comparison?

I like the Excel workbooks with the raw data added to the Github repo, but neither the M. natalensis nor the Lassa tables report month or season of each data point. If these data exist, they should be reported and, ideally, included in the regression model for the

reservoir and pathogen layers. At a very minimum, I would like the authors summarize the seasonality of input data in some way to show that there is not some glaring inconsistency whereby a rodent was never sampled at the time of year relevant for the disease in question in a particular area.

2.Concern over the emphasis on SIRS dynamics over SIR with very little support for this decision.

The authors attempt to address these concerns, and in fact, they do a decent job of emphasizing that support is fairly weak for SIRS assumptions in the results and discussion; however, the methods are still entirely focused on an SIRS approach and now inconsistent with the rest of the paper. This discrepancy needs to be addressed – see specific line by line comments below. Additionally, the authors note in their response that cell-mediated immunity is known to play a role in LASV response in humans and that Abs wane with time—this may be true, but those dynamics are still not SIRS. If humans remain immune but seronegative, they should move into a different class that is Ab negative but certainly not susceptible; these dynamics could be modeled, and I’ll emphasize again that information on the age structure of the serological response would be helpful in assessing this.

3.One other point that I noted on my re-read was that the authors spend a lot of time at the end of the paper predicting annual “cases” (e.g. Fig 6). Perhaps I’ve become too steeped in this difference from COVID, but I would advocate for changing the terminology to “infections” rather than “cases” which to me, implies symptoms. If LASV infections in West Africa are anywhere near as frequent as suggested in this paper, then my guess is that the vast majority are asymptomatic or mildly symptomatic and go unnoticed. Given this, it would be better to describe them as cases instead of infections.

Line-by-line comments:

Abstract:

No need to address this in the abstract necessarily, but you mention ‘West Africa’ throughout the manuscript and show a map of the UN-defined region in all of your figures. It would be helpful to formally define this region (or cite a UN source) somewhere in the text so that the geographic extent of analysis does not seem arbitrary.

Additionally, I think the 4 million (+) annual infections from the SIRS model is a fairly unreasonable projection, and I would suggest to leave this finding out of the abstract.

Author Summary:

Is Nigeria truly at risk for emergence of “new strains” of Lassa virus or just at risk for ‘emergence’? The authors do not report any evidence as to what genotypes to expect in one reason vs. another.

Main paper

Lines 148-153: given the short lifespan of M. natalensis and the seasonal dynamics of Lassa, it seems that 5 seronegative rodents might be easy to acquire. What is the comparative seroprevalence in Lassa-positive regions?

Line 163-167: It would be would be helpful to see a PRISMA diagram in the supplement that explains how you compiled your data for each layer: what terms were searched and surveys were excluded at each point in the analysis. You are very clear about the search terms used for the rodent infections—and you include the helpful Workbooks on Github—but less so here for the humans, and I can’t find the raw human data in the repo. What terms were searched and what serosurveys were excluded at each point in the analysis?

Table 3: Edit % Pos. to % Seropositive. Edit table title to “serosurvey” instead of “survey”

Line 201/Figure 2/Line 278: Why do we assume SIRS as default? In the Results, you report under SIR assumptions but not mention is made of this in the Methods.

Line 307: Again, what about the alternative version in which immunity is maintained? Also, as mentioned above transition to a cell-mediated immune class should not give an SIRS-like dynamic, as individuals who wane from the R class will not move back to S

Line 340: Again, you emphasize this rate seroreversion extensively. I would suggest presenting the uncertainty in this rate—and the two different models derived from that uncertainty at the beginning of this methods section to make it clear that two possible out comes (and a range in between them) are present.

Line 372: As mentioned above, I would like to know if season of sampling for rodents influences the Lassa risk map. Why do you think Lassa is restricted in the west and east if not as a result of human density? Can you discuss this in the Discussion.

Fig 4. What is shown on the x-axis? Probability of a given pixel being LASV (+)? If so, label as such.

Fig 5. You only show serosurveys with pop sizes greater than 50 – why is this? According to Table 3, it looks like some of these surveys must have had very few individuals tested – how did you account for this in your model? This is not clear from the supplement or the main text.

Line 388: As mention above, I would suggest trading “cases” (implying symptomatic cases) with “infections” that might go unnoticed and explain some of these results. This terminology persists throughout the following paragraph

Table 4: Suddenly, all assumptions here switch to SIR when previously the paper emphasized SIRS. This distinction needs to be clarified and consistent throughout. I would suggest presenting results for SIR assumptions only and then including SIRS results in the supplement

**Have all data underlying the figures and results presented in the manuscript been provided?**

Reviewer #1: **No: **Human data for seroprevalence assays appears to be missing from the github repo, or at least difficult to find, as compared with rodent and LASV data.

PLOS authors have the option to publish the peer review history of their article (what does this mean?). If published, this will include your full peer review and any attached files.

Reviewer #1: No
---

## [Editor Report · Decision Letter 2]

17 Feb 2021

Dear Basinski,

We are pleased to inform you that your manuscript 'Bridging the gap: Using reservoir ecology and human serosurveys to estimate Lassa virus spillover in West Africa' has been provisionally accepted for publication in PLOS Computational Biology.

Best regards,

Amy Wesolowski

Associate Editor

PLOS Computational Biology

Nina Fefferman

Deputy Editor

PLOS Computational Biology

---

## [Editor Report · Acceptance letter]

26 Feb 2021

PCOMPBIOL-D-20-01255R2 

Bridging the gap: Using reservoir ecology and human serosurveys to estimate Lassa virus spillover in West Africa

Dear Dr Basinski,

I am pleased to inform you that your manuscript has been formally accepted for publication in PLOS Computational Biology. Your manuscript is now with our production department and you will be notified of the publication date in due course.

With kind regards,

Alice Ellingham
